# Promotion Effect of Agricultural Production Trusteeship on High-Quality Production of Grain—Evidence from the Perspective of Farm Households

Xiaoyan Sun [1], Youchao Wang [1], Fengying Zhu [2], Xiaoyu Liu [3,*], Jianxu Liu [1] and Songsak Sriboonchitta [4]

1   School of Economics, Shandong University of Finance and Economics, Jinan 250014, China; 20080164@sdufe.edu.cn (X.S.); 222101025@mail.sdufe.edu.cn (Y.W.); 20180881@sdufe.edu.cn (J.L.)
2   School of Foreign Languages, Shandong University of Finance and Economics, Jinan 250014, China; 19997902@sdufe.edu.cn
3   Institute of Finance and Economics, Shanghai University of Finance and Economics, Shanghai 200433, China
4   Faculty of Economics, Chiang Mai University, Chiang Mai 50200, Thailand; songsakecon@gmail.com
*   Correspondence: xiaoyu_liu@163.sufe.edu.cn; Tel.: +86-178-6292-2169

**Abstract:** Based on the survey data of five large grain-producing provinces in China, we have studied the promotion effect of the agricultural production trusteeship on the high-quality production of grain by using a propensity score matching method. The empirical results show that the high-quality production of the grain level increases by 0.292, with an increase of 87.4% after farm households participate in agricultural production trusteeship. The level of high efficiency, premiumization, greenization, and branding of grain production increased by 0.234, 0.373, 0.208, and 0.158, respectively. However, there are differences in the facilitation effects of different trusteeship services, with the best promotion effect of agricultural material supply services, followed by post-harvest services, and then land management services and cultivating and harvesting services. The average treatment effect on the treated (*ATT*) is 0.287, 0.230, 0.158, and 0.139, respectively. Meanwhile, there are differences in the promotion effects for farm households with different factor endowments. The promotion effect is better for small farm households with three laborers or less, a land management scale of 10 mus or less, and one type of agricultural machinery or less. Therefore, in order to give full play to the promotion effect of agricultural production trusteeship on high-quality production of grain, the government should vigorously support its development and guide more smallholders to choose agricultural production trusteeship.

**Keywords:** agricultural production trusteeship; high-quality production of grain; propensity score matching

## 1. Introduction

Grain security is "the top concern for the country" [1,2]. Grain security in the new era has a rich connotation, which requires a practical promotion of the high-quality production of grain to achieve the goals of quantitative, qualitative, and ecological grain security in a synergistic manner. At present, China's grain production is relatively safe in terms of quantity, but the grain quality and the ecological environment are worrisome. To this end, the Chinese government has repeatedly stressed the high-quality development of the grain industry, and the grain authorities have also vigorously taken action for high-quality grain production. However, China's grain production is still suffering from problems of the aging, part-time and small-scale operation of farm households, resulting in insufficient ability or high costs for the grain growers to engage in the high-quality production of grain [3–6]. Meanwhile, most of the technical personnel in China's grain industry are gathered in the field of scientific research or in circulation links, with a relative shortage of professionals in the production sector. However, the high-quality production of grain needs professional

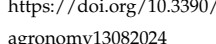



support in the production sector. Researchers have shown that agricultural production services can offer a large number of professional and technical personnel and advanced agricultural machinery for grain production [7], which can not only solve the technical and cost challenges that farm households face in grain production [8,9] but also help reduce the use of chemical fertilizers and pesticides [10–12]. The above studies affirm the role of agricultural production services in promoting the high efficiency and greenization of grain production. Meanwhile, there is still a need for more systematic and in-depth research on the possibility of agricultural productive services for promoting the high-quality production of grain at multiple levels in a coordinated manner.

Agricultural production trusteeship is a full-process, specialized, and large-scale form of agricultural production service. By the end of 2021, China's agricultural production trusteeship service area had exceeded 1.67 billion mu, making it the most important agricultural production service form in China. In China, the statistical standard for land area is usually "mu" (1 mu = 666.67 $m^2$ = 1/5 $hm^2$), so the land area in this article is expressed in mu), covering over 78 million small farm households (See China's Ministry of Agriculture and Rural Affairs—Bearing Steady the Heavy Responsibility of Grain Production for Consolidating the Foundation of a Strong Nation. http://www.ghs.moa.gov.cn/xczx/2022 10/t20221021_6413824.htm (accessed on 10 March 2023)). Hence, the purpose of this paper is to carry out an in-depth and systematic study of the promotion effect of agricultural production trusteeship on the high-quality production of grain in multiple dimensions including high efficiency, premiumization, greenization, and branding of grain production, so as to better accomplish the multidimensional goals of China's grain quantity security, quality security, and ecological security.

The chapters are organized as follows: First, a literature review and theoretical framework is presented. This part primarily reviews the connotation of the high-quality production of grain and the relationship between the high-quality production of grain and agricultural production trusteeship. Next, the data sources, research methods, and variable selection of this paper are introduced. Then, the empirical study is conducted. This section includes the overall promotion and heterogeneous effects of agricultural production trusteeship on the high-quality production of grain.

## 2. Literature Review and Theoretical Framework

### 2.1. Literature Review

2.1.1. Connotation and Dimension of High-Quality Production of Grain

Judging from the available literature, there is not much research on the high-quality production of grain at present, most of which focuses on the high-quality development of agriculture and the grain industry. Some researchers argue that the high-quality development of agriculture should combine quality and efficiency promotion, greenization, and sustainable development, so as to achieve the goals of "prospering agriculture by efficiency", "prospering agriculture by quality", and "prospering agriculture by greenization" [13,14].

The high-quality development of the grain industry is a proper part of the high-quality development of agriculture [15]. Hence, many researchers carry out research on this subject. Drawing on the dimensions of high-quality economic development, Huifen Chen et al. studied the high-quality development of the grain industry from the five dimensions of high efficiency, premiumization, coordination, greenization, and internationalization [16]. Di Qi et al. proposed a method of high-quality development of the grain industry based on the target of multidimensional security objectives of grain, with regard to quality, efficiency, structure, and the environment [17]. Xiang Fei et al. conducted research from the perspective of the grain industry chain, with regard to product quality, circulation efficiency, large-market construction, technological support for the industry chain, and stability of industrial policies [15].

### 2.1.2. Relationship between Agricultural Production Trusteeship and High-Quality Production of Grain

Agricultural production trusteeship refers to a mode of agricultural operation in which farm households and other management entities entrust all or part of the operational links in agricultural production to agricultural production service organizations for completion or assistance without transferring their land management rights [18]. This can effectively promote the quality, efficiency, and greenization of grain production. With regard to the improvement of quality and efficiency, researchers have found that agricultural production services can improve the efficiency of the use of resource factors by optimizing factor inputs and can thereby promote sustainable grain production [19]. Agricultural technology promotion services, agricultural information services, and agricultural product marketing services are conducive to improving the willingness and ability of farm households to produce and sell high-quality agricultural products [20,21]. Moreover, the production link outsourcing services help agricultural operators to have extra financial resources and energy to create brands for quality agricultural products [22,23]. With regard to green production, researchers have found that agricultural production services can contribute to the increase in green productivity in agriculture by motivating farm households to adopt green agricultural production techniques and to reduce the use of pesticides and chemical fertilizers [24,25]. At the same time, they can break through the restriction of farm households' low willingness to choose green production and directly drive farm households who have been offered agricultural production services to practice green production.

### 2.1.3. Shortcomings of the Existing Literature and the Main Contributions of This Paper

Although some researchers affirmed the promotion effect of various types of agricultural production services on grain efficiency and greenization production, there are still some shortcomings. For example, there is a lack of research on the connotation of high-quality production of grain. At the same time, most of the available literature focuses on the effect of individual services (e.g., green prevention and control techniques, soil testing formula fertilization, etc.) on grain production [26–29], or on the impact of agricultural production services on the individual dimensions of grain production (e.g., production efficiency, greenization, etc.) [30].

This paper will conduct a comprehensive and systematic study. The main contributions are the following: Firstly, we will define the connotation of high-quality production of grain, then we will construct the indexes of the high-quality production of grain from the four dimensions of high efficiency, premiumization, greenization, and branding, in order to comprehensively measure the level of high-quality production of grain. Secondly, we will select a one-stop and all-inclusive form of agricultural production services, the agricultural production trusteeship, to study the coordinated promotion effect of the comprehensive service on the high-quality production of grain. Furthermore, we will examine the heterogeneity of the promotion effect of agricultural production trusteeship on the high-quality production of grain from high-quality production dimensions, the trusteeship service links, and the categories of farm households.

### 2.2. Theoretical Framework

By drawing the connotation of high-quality development of agriculture and the grain industry [15,31,32], we argue that the high-quality production of grain is a way of producing and operating grain that is based on an assurance of efficient grain production. It includes high efficiency, premiumization, greenization, and branding of grain production. High-quality grain production is the foundation of ensuring China's grain security in the new stage. The whole process of professional and large-scale agricultural production trusteeship services can effectively promote the high efficiency, premiumization, greenization, and branding of grain production, and then strongly promote high-quality grain production, and effectively guarantee the grain quantity, quality, and ecological security of China from the production end. The specific framework for agricultural production trusteeship to

promote high-quality production of grain is shown in Figure 1 (Figure 1 is produced by the authors based on the theoretical framework).

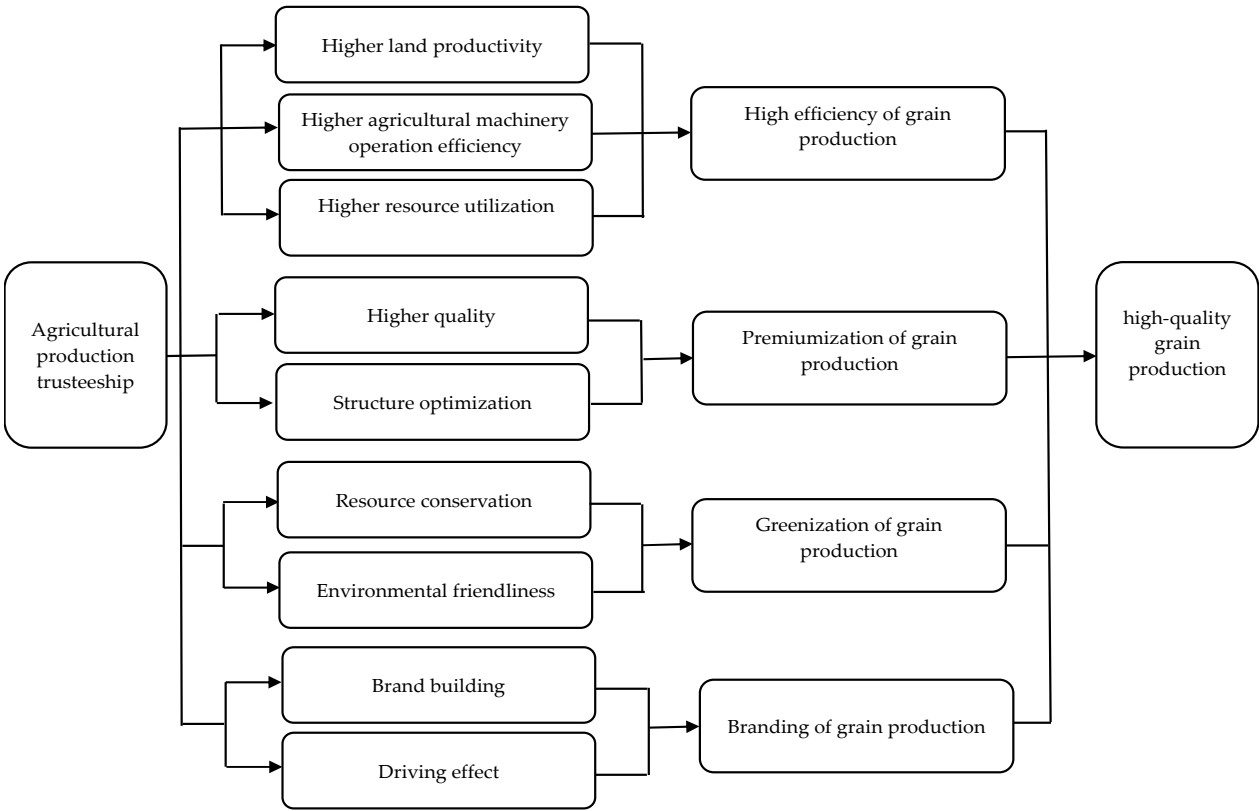

**Figure 1.** Mechanism of agricultural production trusteeship in promoting high-quality production of grain.

### 2.2.1. Contributing to the High Efficiency of Grain Production

Firstly, agricultural production trusteeships can improve the efficiency of land output. On the one hand, agricultural production trusteeship can free up more arable land by abolishing field ridges and reorganizing roads, and the utilization rate of arable land can be improved; on the other hand, the use of good seeds of grain and intensive, standardized, and scientific planting services are also conducive to the increase in grain yield [33–35]. Secondly, it can improve the efficiency of the use of agricultural machinery. Compared to the small-scale scattered operation of professional agricultural machinery households, agricultural production trusteeship is for large-scale operation on large, connected patches of land, which significantly reduces losses of agricultural machinery when transferring from one patch of land to another. Furthermore, a large-scale operation is also conducive to the improvement of the operation efficiency of the original small agricultural machinery and the use of high-efficiency large agricultural machinery [30]. Thirdly, it can improve resource utilization efficiency. Intensive, standardized, and scientific cultivation services are conducive to improving the use efficiency of chemical fertilizers, pesticides, and other inputs [19,36]. Agricultural production trusteeship promotes the high efficiency of grain production by improving the output efficiency of various production factors.

### 2.2.2. Contributing to the Premiumization of Grain Production

On the one hand, agricultural production trusteeship is conducive to the improvement of quality. It can greatly improve grain quality through the use of good seeds, green cultivation, and scientific management [37]. Meanwhile, the post-production drying and storage services help to reduce the degree of post-harvest decline of the grain quality, thus improving the quality of grain along the whole chain [38]. On the other hand, it is

conducive to structural optimization. The agricultural production trusteeship organization can better grasp the market demand, produce the high-quality grain and special grain that is urgently needed by the market, and avoid the oversupply of common grain, thus the grain structure can be optimized. As can be seen, agricultural production trusteeships can contribute to the premiumization of grain production in terms of both grain quality improvement and structural optimization.

### 2.2.3. Contributing to the Greenization of Grain Production

Firstly, agricultural production trusteeships can promote resource conservation. The use of chemical fertilizers, pesticides, and seeds can be effectively reduced by applying technologies such as soil testing formula fertilization, integrated prevention and control, and seed coating [10–12]. Meanwhile, the application of sprinkler and drip irrigation technology can effectively save water resources. Secondly, it is beneficial to environmental friendliness. The agricultural production trusteeship service usually adopts low-toxic and environmentally friendly pesticides, fertilizers, and other green inputs, and adopts green production technologies such as integrated prevention and control and straw return. These green production behaviors can not only directly reduce environmental pollution in the process of grain production but also help improve farm households' green cognition level and improve their adoption of green production technologies, thereby effectively reducing the possible non-point source pollution caused by grain cultivation [39,40]. Through resource conservation and environmental friendliness, agricultural production trusteeships can thus promote the greenization of grain production.

### 2.2.4. Contributing to the Branding of Grain Production

Brand building is inseparable from brand awareness and scale foundation, but small farm households hardly have the awareness and capability of establishing a brand. Even if some new operation entities have certain brand awareness, it is difficult for them to build high-quality grain brands due to their small operation scale, shortage of funds, and lack of publicity channels. Agricultural production trusteeship service organizations can give play to its scale advantage, organizational advantage, and professional advantage and organize farm households to produce high-quality grain on one end and build brands on the other. Through building high-quality grain brands, agricultural production trusteeship organizations can help achieve high quality and good prices of grain in order to protect the income of farm households and stimulate their motivation to grow grain.

## 3. Data Sources, Research Methods, and Variable Selection
### 3.1. Data Sources

The data used in this paper were obtained through a questionnaire survey conducted between October and December 2020 among the farm households in Henan, Shandong, Anhui, Jiangsu, and Hebei provinces in China. The selection of the provinces for the survey is primarily based on two considerations: Firstly, the wheat cultivation area is wide and the yield is higher; secondly, the agricultural production trusteeship services are carried out earlier and are more mature. Therefore, the five provinces, including Henan Province and Shandong Province, which are the major wheat cultivation provinces in China and where the agricultural production trusteeship services are well developed were selected. Henan, Shandong, Anhui, Jiangsu, and Hebei are the top 5 provinces in the country in terms of wheat production. In 2019, the wheat sowing area and total production of these 5 provinces were 17,213.43 thousand hectares and 107,316.6 thousand tons, respectively, accounting for 72.55% and 80.33% of the national wheat sowing area and national production, respectively. By the end of 2019, the area under agricultural production trusts in Hebei, Anhui, Shandong, and Henan provinces had all exceeded 100 million mus or more, and the area under agricultural production trusts in Jiangsu province had reached more than 50 million mus. Before the formal investigation, our research team carried out a preliminary investigation in a number of typical administrative villages in Shandong Province. Our investigation

team consists of 25 people, including 5 teachers from Shandong University of Finance and Economics, 3 doctoral students, 12 graduate students, and 5 undergraduates. The team was divided into five groups, each led by a teacher and responsible for the investigation in one province. The members of the investigation team received professional questionnaire training before conducting the formal survey, and the training methods include practical training and conference training. Through training, team members were very familiar with the content of the questionnaire and had a deep understanding of the methods and other points of attention of the questionnaire. Through the pre-survey, we modified and improved the survey questionnaire, so as to determine the formal survey questionnaire. In accordance with the sampling criteria that the study sites should represent different regions and different levels of economic development, in the formal survey, we used a combination of stratified and random sampling to select 3 counties (or cities or districts) in each province that carry out agricultural production trusteeship. Then from each of the counties (or cities or districts), 3 townships (or towns) where agricultural production trusteeship services were carried out were further selected, and then from each of the 3 townships (or towns), 2 administrative villages where agricultural production trusteeship services were carried out were further selected, and finally, from each of the administrative villages, 10–20 farm households were randomly selected. The research was conducted by means of one-to-one interviews with the farm households, with questions in the questionnaires asked and questionnaires filled in personally by us. The questions included in the questionnaires are related to individual characteristics, family characteristics, grain production characteristics, social relationship characteristics, etc. After removing questionnaires that omitted key information, had missing variables, had contradictory information, or were outliers, a total of 1174 valid questionnaires were obtained, with an effective rate of 93.92%.

*3.2. Research Methodology*

On the one hand, farm households' choice of agricultural production trusteeship is a non-random "self-choosing" behavior, which will be influenced by the individual, family, operation, and social characteristics of the decision makers, and these factors will also affect the quality of their grain production. Therefore, the research on the promotion effect of agricultural production trusteeship on the high-quality production of grain has endogenous problems, which make it impossible to evaluate the effect accurately. On the other hand, although it is possible to observe each farm household's grain production situation before and after his choice of agricultural production trusteeship, each control variable of sample farm households may vary before and after their choice of agricultural production trusteeship, resulting in the variables being uncontrolled. Therefore, we can only simulate the situation when the farm households do not participate in agricultural production trusteeship and then compare the simulated situation when the farm households do not participate in agricultural production trusteeship with the situation in which they do. To solve the above-mentioned problems, we have adopted the Propensity Score Matching method (PSM) [41,42] in this research to test the promotion effect of agricultural production trusteeship on the high-quality production of grain.

First, farm households that are now under agricultural production trusteeship (the experimental group) and those who are not under agricultural production trusteeship (the control group) were separated according to observable variables. To this end, we used the fitting value of the conditional probability of the farm households when they choose agricultural production trusteeship as the propensity score, which is estimated using the Logit model:

$$P(X_i) = Prob(D_i = 1 | X_i) \tag{1}$$

where *i* denotes an individual farm household, $D_i = 1$ denotes the farm household *i*'s choice of agricultural production trusteeship, and $X_i$ denotes a series of control variables that may affect the farm household i's choice of agricultural production trusteeship. After estimating the propensity score, the experimental and control groups are matched by constructing a counterfactual framework. A good matching estimator requires a large common support

domain for the propensity score of the farm households who have chosen agricultural production trusteeships and those who have not after matching. Meanwhile, to ensure the robustness of the matching results, we use k-nearest neighbor matching, caliper matching, kernel matching, local linear regression matching, and spline matching for estimation. The effect of agricultural production trusteeship on promoting the high-quality production of grain can be expressed by the Average Treatment effect (*ATT*) of the farm households participating in agricultural production trusteeship, i.e.,

$$ATT = E(y_{1i} - y_{0i} | D_i = 1) \tag{2}$$

where $y_{1i}$ denotes the high-quality production of grain situation when the farm household *i* is under agricultural production trusteeship and $y_{0i}$ denotes the high-quality production of grain situation when it is not under agricultural production trusteeship after matching. Due to the fact that the grain production situations of the farm households who are now under agricultural production trusteeship before they are under agricultural production trusteeship cannot be measured, we simulate the characteristics of the farm households in the control group through various matching methods to ensure the matched experimental and control groups of the farm households have no significant differences in characteristics in other situations than in those that they are or not under agricultural production trusteeship. Then it can be taken that we can simultaneously observe the high-quality production of grain situations of the same farm household before and after it is under agricultural production trusteeship.

*3.3. Variable Selection and Descriptive Statistics*

3.3.1. Explained Variable

We have explained the high-quality production of grain situations from the perspective of farm households, which can better describe the production process of grain and can avoid the limitations of examining high-quality production in terms of finished grain products only. Hence, the explained variable is the level of high-quality production of grain in farm households, which is measured by the index of high-quality production of grain. On the basis of the above definition of high-quality production of grain, we design the evaluation index system for the high-quality production of grain in the four dimensions of high efficiency, premiumization, greenization, and branding, as shown in Table 1.

High efficiency is the primary prerequisite for the high-quality development of grain production. The most direct manifestation of high efficiency is the improvement of the utilization efficiency of various production factors [19,30,33–36]. Hence, agricultural machinery efficiency, land production efficiency, and resource utilization efficiency are adopted in this research to reflect the high efficiency of grain production.

Premiumization is an essential requirement of the high-quality production of grain. As consumer demand has escalated, food consumption has shifted from having enough to eat to eating better, which naturally requires a quality food supply. A premium grain supply requires both the improvement of the grain quality and the optimization of the grain structure [37,38]. To meet the above requirements, it is necessary to select high-quality grain varieties and standardize their cultivation. Hence, we adopt the indexes of the two elements of the use of high-quality seeds and standardized production to measure the premiumization of grain production.

Greenization forms an important part of the high-quality production of grain. The goal of green production is resource conservation and environmental friendliness [10–12,43], which requires the use of green inputs and green agricultural production technologies. Hence, we adopt the above two indexes to measure the greenization of grain production.

**Table 1.** Index system for evaluation of farm households' high-quality grain production level and its weights.

| Dimension Indexes | Element Indexes | Measurement Method |
|---|---|---|
| High efficiency of grain production (0.15308) | Efficiency of farm machinery operation (0.05696) | Relative to average level of operation, the agricultural operators' assessment of the operational efficiency of the agricultural machinery they use: very low = 1, relatively low = 2, fairly average = 3, a little bit higher = 4, much higher = 5 |
| | Efficiency of land output (0.04942) | Yield of wheat per mu in 2020 (kg/mu) |
| | Efficiency of resource utilization (0.04669) | Relative to average use, the agricultural operators' assessment of the amount of agricultural inputs used: much more = 1, more = 2, fairly average = 3, less = 4, much less = 5 |
| Premiumization of grain production (0.49035) | Use of high-quality seeds (0.32074) | Whether high-quality seeds are used: yes = 1, no = 0 |
| | Standardized production (0.16961) | The number of the four links of unified seed supply, unified farm machinery operation, unified land management and unified marketing which the farm households have used: 0,1, 2, 3 or 4 |
| Greenization of grain production (0.32883) | Use of green inputs (0.27415) | Types of green pesticides (including biopesticides, pollution-free pesticides, etc.) and green fertilizers (including organic fertilizers, green fertilizers, etc.) used: 0, 1 or 2 |
| | Use of green production technology (0.05468) | The number of green agricultural production techniques of seed coating, deep tillage and subsoiling, soil testing formula fertilization, green prevention and control techniques, sprinkler and drip irrigation, and straw returning used: 0, 1, 2, 3, 4, 5 or 6 |
| Branding of grain production (0.02774) | High quality and good price mechanism (0.02774) | The agricultural operator's assessment of the price of wheat sold relative to the market price: low = 1, slightly lower = 2, fairly average = 3, slightly higher = 4, much higher = 5 |

Branding is the ultimate expression and guarantee of high-quality production of grain. High-quality grain can enhance the product image through product certification and brand building, thereby realizing high quality and good prices of grain and forming a positive interaction between production and marketing. Therefore, we have adopted "the achievement or not of high quality and good prices of grain" to reflect the branding of grain.

Drawing on the available research [44,45], we adopt the entropy weight method to measure the high-quality production of grain level from the four dimensions of high efficiency, premiumization, greenization, and branding.

Firstly, the indexes of the four dimensions of high efficiency, premiumization, greenization, and branding are standardized by the following formula:

$$y_{ij}^{nor} = \frac{y_{ij} - y_{ij}^{min}}{y_{ij}^{max} - y_{ij}^{min}} \tag{3}$$

where $y_{ij}^{nor}$ refers to the standardized index of each dimension, $i$ denotes the individual farm household, $j$ ($j$ = 1, 2, 3, 4) denotes the four dimensions of high-quality grain production, $y_{ij}^{max}$ is the maximum value of the $j$-th dimension of the farm household $i$, and $y_{ij}^{min}$ is the minimum value of the $j$-th dimension of the farm household $i$. High efficiency, premiumization, greenization, and branding are all positively related to high-quality grain production, and thus the indexes of the four dimensions have directionally consistent measures without the need for inverse treatment.

Then, the information entropy of the indexes of the four dimensions of high efficiency, premiumization, greenization, and branding is calculated with the following formula:

$$E_j = -\frac{1}{\ln n} \sum_{i=1}^{n} p_{ij} \ln p_{ij} \tag{4}$$

where $p_{ij} = y_{ij}^{nor} / \sum_{i=1}^{n} y_{ij}^{nor}$, and if $p_{ij} = 0$, then $\lim_{p_{ij} \to 0} p_{ij} \ln p_{ij} = 0$.

Then, the weight of the indexes of the four dimensions of high efficiency, premiumization, greenization, and branding is calculated with the following formula:

$$W_j = \frac{1 - E_j}{\sum_{j=1}^{4} (1 - E_j)} \tag{5}$$

where specific weights for the indexes of the four dimensions of high efficiency, premiumization, greenization, and branding of grain production are shown in Table 1.

Finally, the comprehensive index of high-quality production of grain is calculated using the following formula:

$$Y_i = \sum_{j=1}^{4} y_{ij}^{nor} W_j \tag{6}$$

### 3.3.2. Core Explanatory Variable

We have focused on the promotion effect of agricultural production trusteeship on the high-quality production of grain. Grain cultivation requires a range of services before, during, and after grain production, including agricultural materials supply service, ploughing service, seeding service, production management service, harvesting service, and post-harvest service. Agricultural production trusteeship organizations can provide the six above-mentioned types of services for grain production in a one-stop, all-inclusive manner. Since most farm households need four services, including agricultural materials supply service, ploughing service, seeding service, and harvesting service, regardless of whether they purchase agricultural production trusteeship services, in this paper, farm households who are under agricultural trusteeship service are defined as those who purchase four

or more kinds of trusteeship services from the same trusteeship organization, while farm households who are not under agricultural production trusteeship are defined as those who purchase less than four kinds of services from the same trusteeship organization.

### 3.3.3. Control Variables

When PSM is used for analysis, it is important to include in the model, as far as possible, the variables that affect both farm households' choice of agricultural production trusteeship and high-quality production of grain. Finally, we select the individual characteristics (Age and Educational background) of the grain-cultivation decision-makers, family characteristics (agricultural labor force, part-time business, total household income, proportion of grain income, and market distance), operation characteristics (scale of operation, degree of land fragmentation, types of agricultural machinery, and traffic conditions), and social characteristics (information acquisition channels, availability of farm machinery household resources, membership of cooperative organizations, government publicity campaigns, and professional cultivation guidance) as control variables that affect farm households' choice of agricultural production trusteeship and high-quality production of grain. See Table 2 for the definition and descriptive statistics of each variable.

**Table 2.** Definition and descriptive statistics of the variables.

| Variable Name | Variable Definition and Assignment | Mean | Standard Deviation |
|---|---|---|---|
| High-quality production of grain | This is the high-quality grain production index, the higher the value of which, the higher the high-quality grain production level | 0.461 | 0.256 |
| High efficiency of grain production | This is the efficient grain production index, the higher the value of which, the higher the grain production efficiency level | 0.527 | 0.174 |
| Premiumization of grain production | This is the premium grain production index, the higher the value of which, the higher the grain production premiumization level | 0.477 | 0.379 |
| Greenization of grain production | This is the green grain production index, the higher the value of which, the higher the grain production greenization level | 0.396 | 0.308 |
| Branding of grain production | This is the branding index of grain production, the higher the value of which, the higher the branding level of grain production | 0.569 | 0.189 |
| Whether participating in the agricultural production trusteeship | Whether or not farm households were under agricultural production trusteeship in 2020: yes = 1, no = 0 | 0.497 | 0.5 |
| Age | This is the age of grain production decision-makers (years) | 56.5 | 10.347 |
| Education background | This is the educational background of grain production decision-makers: primary school and below = 1, junior high school = 2, senior high school = 3, college = 4, undergraduate and above = 5 | 1.814 | 0.792 |
| Number of agricultural labor force | This was the number of household agricultural labor force in 2020 (persons) | 1.884 | 0.557 |
| Part-time business | Whether or not the agricultural labor force was working or doing business during the agricultural leisure time in 2020: yes = 1, no = 0 | 0.653 | 0.476 |
| Total household income | This was the total household income in 2020: less than RMB 50,000 = 1, between RMB 50,000 and RMB 100,000 = 2, RMB 100,000 or above = 3 | 1.973 | 0.76 |
| Proportion of grain income | This is the proportion of grain income in total household income in 2020: below 20% = 1, 20% or between 20% and 50% = 2, 50% or between 50% and 80% = 3, 80% or above = 4 | 2.225 | 1.081 |
| Market distance | This is the distance between farm households and large agricultural markets (km) | 3.959 | 2.514 |
| Scale of Operation | This was the household Area of wheat planted in 2020 (mu) | 21.488 | 151.374 |
| Degree of land fragmentation | This was the number of patches of land/scale of operation in 2020 | 0.476 | 0.489 |

**Table 2.** *Cont.*

| Variable Name | Variable Definition and Assignment | Mean | Standard Deviation |
|---|---|---|---|
| Types of agricultural machinery | This is the household types of farm machinery, including ploughing machinery, subsoiler, seeder (or seed and fertilizer co-sowing machine), fertilizing machinery, motor-driven pesticide application equipment, watering equipment, harvester, agricultural conveyor and thresher, calculated in round-off numbers | 1.383 | 1.437 |
| Traffic conditions | This is the accessibility of large agricultural machinery to the cultivated land: very inconvenient = 1, not very convenient = 2, general = 3, relatively convenient = 4, very convenient = 5 | 4.118 | 0.865 |
| Information acquisition channels | These are the six channels for farm households to get access to agricultural information, including relatives and friends, television, mobile phone or computer, cooperatives and other organizations, village committees or government departments, and agricultural materials distributors, calculated in round-off numbers | 3.423 | 1.301 |
| Availability of farm machinery household resources | Whether or not farm households have farm machinery household resources: yes = 1, no = 0 | 0.794 | 0.405 |
| Membership of cooperative organizations | Whether or not farm households joined organizations such as cooperatives: yes = 1, no = 0 | 0.491 | 0.5 |
| Government publicity situation | This is the number of agricultural production trusteeship publicity campaigns targeted by government departments in a year | 3.428 | 1.921 |
| Professional cultivation guidance | This is the number of on-site cultivation instruction by professional technicians within a year | 2.501 | 2.073 |

Note: The variables Age, Market distance, and Scale of operation have been logarithmized in the regressions in order to reduce the effect of the dimension.

## 4. Empirical Results and Analysis

### 4.1. Estimation of the Decision-Making Model of the Farm Households to Choose Agricultural Production Trusteeship

The probability of farm households' choice of agricultural production trusteeship is estimated using the Logit model. The estimation results are shown in Table 3. The variables of individual characteristics, family characteristics, management characteristics, and social characteristics of the grain decision makers have a significant influence on the possibility of their choice of agricultural production trusteeship. Among them, age, educational background, part-time business, total family income, proportion of grain income, market distance, traffic conditions, information acquisition channels, membership of cooperative organizations, government publicity campaigns, and professional cultivation guidance have significant positive effects on farm households' choice of agricultural production trusteeship. The variables of the agricultural labor force, degree of land fragmentation, types of agricultural machinery, and availability of farm machinery household resources have a significant negative effect on farm households' choice of agricultural production trusteeship.

**Table 3.** Logit model of farm households' decision-making in choosing agricultural production trusteeship.

| Variable Name | Estimated Value of Coefficient | Standard Deviation | Z Value |
|---|---|---|---|
| Age | 3.068 *** | 0.507 | 6.05 |
| Education background | 0.327 *** | 0.1 | 3.27 |
| Number of agricultural labor force | −0.515 *** | 0.143 | −3.61 |
| Part-time business | 0.652 *** | 0.197 | 3.3 |
| Total household income | 0.719 *** | 0.119 | 6.06 |
| Proportion of grain income | 0.177 ** | 0.083 | 2.12 |
| Market distance | 0.305 * | 0.168 | 1.82 |
| Scale of operating | −0.056 | 0.12 | −0.47 |
| Degree of land fragmentation | −0.667 *** | 0.226 | −2.95 |
| Type of agricultural machinery | −0.313 *** | 0.061 | −5.18 |
| Traffic conditions | 0.556 *** | 0.116 | 4.81 |
| Channels of information acquisition | 0.241 *** | 0.083 | 2.91 |
| Availability of farm machinery household resources | −0.418 ** | 0.179 | −2.33 |
| Membership of cooperative organizations | 0.470 ** | 0.204 | 2.3 |
| Government publicity campaigns | 0.324 *** | 0.044 | 7.38 |
| Professional cultivation guidance | 0.144 *** | 0.039 | 3.68 |
| Constant term | −18.370 *** | 2.303 | −7.98 |
| LR statistics | 483.397 *** | | |
| Pseudo $R^2$ | 0.297 | | |
| Sample size | 1174 | | |

Note: The results of k-nearest neighbor matching (k = 4) are listed in this table and the results of other matching methods are similar and so are not listed here. *, ** and *** indicate that it is significant at the statistical levels of 10%, 5%, and 1%, respectively.

### 4.2. Common Support Domain and Balance Test

#### 4.2.1. Common Support Domain

Ensuring matching quality requires a common support domain check [46]. As shown in Figure 2, the probability distribution of propensity scores of the farm households in the experimental group and those in the control group varies greatly before matching, with the experimental group concentrated in the higher value domain, while the control group is concentrated in the lower value domain. After matching, the difference in the probability distribution of propensity scores of the two groups of farm households is obviously reduced, and the common support domain is larger. Referring to the above data tables, 28 samples are lost in the experimental group and 80 samples are lost in the control group, with approximately 90.8% of the samples in the common support domain, so it is a good match.

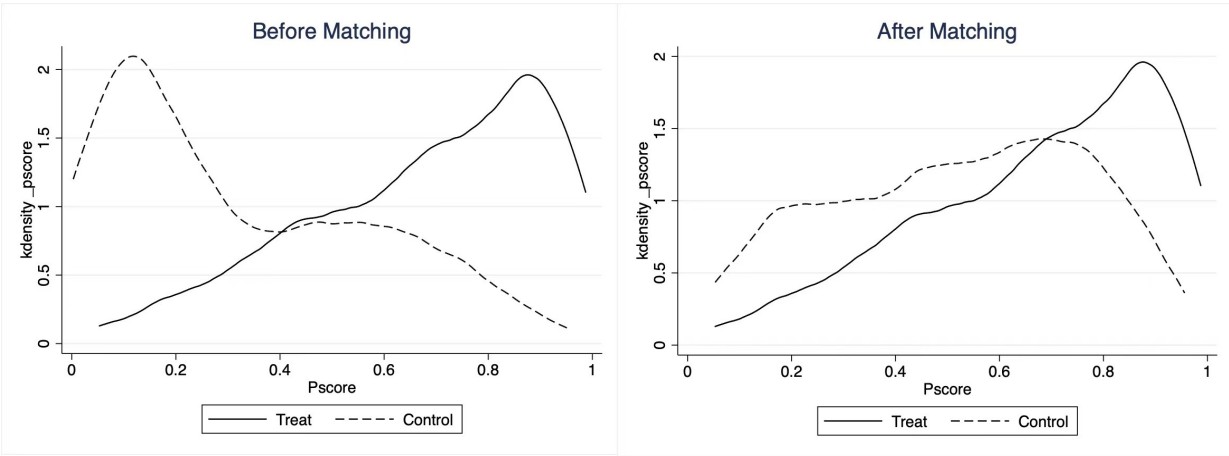

**Figure 2.** Density function before and after propensity score matching.

4.2.2. Balance Test

After matching, it is necessary to test for significant differences in the control variables between the experimental group and the control group of farm households. This is called a balance test [46]. As shown in Table 4, the Pseudo $R^2$ value significantly decreases from 0.299 before matching to 0.013 after matching; the LR statistic significantly decreases from 486.15 to 20.2–20.7, and the joint significance test of control variables changes from the 1% significance level to high-probability rejection, with the mean deviation and median deviation decreasing to within 10%. As a result, the total deviation after matching is significantly lower and the matching result is more satisfactory.

**Table 4.** Balance test of control variables before and after matching.

| Matching Methods | Pseudo $R^2$ | LR Statistics | *p*-Value | Mean Deviation (%) | Median Deviation (%) |
|---|---|---|---|---|---|
| Before matching | 0.299 | 486.15 | 0.000 | 35.3 | 28.5 |
| K-nearest neighbor matching | 0.013 | 20.7 | 0.19 | 4.9 | 5.4 |
| Caliper matching | 0.013 | 20.2 | 0.211 | 3.9 | 3.1 |
| Core matching | 0.013 | 20.33 | 0.206 | 4.1 | 3.8 |
| Local linear regression matching | 0.013 | 20.35 | 0.205 | 4.3 | 3.6 |
| Spline matching | 0.013 | 20.35 | 0.205 | 4.3 | 3.6 |

Note: This table shows the results of the balance test for the control variable of the equation of the high-quality grain production index. The balance test results of other equations are similar and are not reported in this case.

*4.3. Overall Promotion Effect of Agricultural Production Trusteeship on High-Quality Production of Grain*

To ensure the robustness of the regression results, five matching methods, i.e., k-nearest neighbor matching, caliper matching, kernel matching, local linear regression matching, and spline matching, are used in this paper to estimate the promotion effect of agricultural production trusteeship on the high-quality production of grain. As shown in Table 5, the five estimates are highly similar, with the average treatment effect on the treated (*ATT*) passing the test at the significance level of 1%. According to the *ATT* average, the high-quality production of grain level of farm households is 0.292 higher (an increase of 87.4%) after they are under agricultural production trusteeship compared to the level that would have been achieved if they had not been under agricultural production trusteeship. Among them, the levels of high efficiency, premiumization, greenization, and branding increased by 0.234, 0.373, 0.208, and 0.158, respectively, with an increase of 54.8%, 119.9%, 66.7%, and 31.7%, respectively. This thus proved that agricultural production trusteeship can promote the high efficiency, premiumization, greenization, and branding of the grain production of the farm households, which then promotes the high-quality production of grain in multiple dimensions in a synergistic manner.

**Table 5.** Overall promotion effect of agricultural production trusteeship on high-quality grain production.

| Matching Method | K-Nearest Neighbor Matching | Caliper Matching | Core Matching | Local Linearity Regression Matching | Spline Matching | Average |
|---|---|---|---|---|---|---|
| High-quality grain production | | | | | | |
| Experimental group | 0.626 | 0.626 | 0.626 | 0.626 | 0.626 | 0.626 |
| Control group | 0.327 | 0.334 | 0.335 | 0.334 | 0.34 | 0.334 |
| *ATT* | 0.3 *** | 0.292 *** | 0.291 *** | 0.292 *** | 0.286 *** | |
| | (0.022) | (0.02) | (0.02) | (0.025) | (0.02) | 0.292 |
| High efficiency of grain production | | | | | | |
| Experimental group | 0.661 | 0.661 | 0.661 | 0.661 | 0.661 | 0.661 |
| Control group | 0.425 | 0.426 | 0.427 | 0.425 | 0.431 | 0.427 |
| *ATT* | 0.236 *** | 0.235 *** | 0.234 *** | 0.237 *** | 0.23 *** | |
| | (0.011) | (0.011) | (0.011) | (0.013) | (0.01) | 0.234 |
| Premiumization of grain production | | | | | | |
| Experimental group | 0.684 | 0.684 | 0.684 | 0.684 | 0.684 | 0.684 |
| Control group | 0.297 | 0.312 | 0.314 | 0.311 | 0.32 | 0.311 |
| *ATT* | 0.388 *** | 0.372 *** | 0.37 *** | 0.373 *** | 0.364 *** | |
| | (0.037) | (0.034) | (0.033) | (0.042) | (0.033) | 0.373 |
| Greenization of grain production | | | | | | |
| Experimental group | 0.52 | 0.52 | 0.52 | 0.52 | 0.52 | 0.52 |
| Control group | 0.311 | 0.31 | 0.311 | 0.313 | 0.314 | 0.312 |
| *ATT* | 0.209 *** | 0.21 *** | 0.21 *** | 0.207 *** | 0.206 *** | |
| | (0.029) | (0.026) | (0.026) | (0.034) | (0.023) | 0.208 |
| Branding of grain production | | | | | | |
| Experimental group | 0.657 | 0.657 | 0.657 | 0.657 | 0.657 | 0.657 |
| Control group | 0.498 | 0.499 | 0.499 | 0.498 | 0.503 | 0.499 |
| *ATT* | 0.159 *** | 0.158 *** | 0.158 *** | 0.159 *** | 0.153 *** | |
| | (0.014) | (0.014) | (0.013) | (0.017) | (0.014) | 0.158 |

Note: *** indicate that it is significant at the statistical level of 1%; the standard deviations are shown in brackets.

The conclusion of this paper is in good agreement with the available research. With regard to high efficiency, Long Qian et al. suggest that large-scale agricultural machinery services can improve labor productivity [47]; the research by Yajuan Li also argues that socialized services can improve the utilization efficiency of chemical fertilizer resources [19]; meanwhile, Tao Chen and Siyu Yang suggest agricultural materials supply and agricultural machinery services can significantly and positively affect the yield of rice and wheat [7,30]. In this paper, we find that agricultural production trusteeships can promote the high efficiency of grain production in a coordinated manner from three aspects: The utilization efficiency of agricultural materials, agricultural machinery, and grain yield. With regard to premiumization, we find that agricultural production trusteeship can synergistically promote the premiumization of grain production in terms of both the use of good varieties and standardized services. Moreover, the research by Michael J Edney also suggests that the adoption of good seeds can significantly improve the quality of wheat [37]; the research by Rani Puthukulangara Ramachandran also shows that post-harvest storage, transportation, and marketing services can prevent food spoilage [38]. With regard to greenization, Chunfang Yang argues that the socialized service of green production significantly reduces the fertilizer application level of rice farmers in Jiangsu Province of China [11]; the research by Zhong Ren suggests that the socialized services of agriculture significantly and positively affect the application behavior of organic fertilizer of the farm households in Shandong Province of China, and the application of organic fertilizer can reduce agricultural non-point source pollution significantly [10].

It is thus obvious that this research maintains high consistency with previous studies yet is more in-depth and comprehensive. In this paper, we not only study the impact of individual services on individual aspects of high-quality production of grain but also construct a comprehensive system of indexes to study the synergistic promotion effect of production trusteeship services on the high-quality production of grain in a multi-dimensional and multi-faceted way.

### 4.4. Heterogeneity Analysis of Agricultural Production Trusteeship in Promoting High-Quality Production of Grain

4.4.1. Heterogeneity of Different Trusteeship Services in Promoting High-Quality Grain Production

As mentioned above, agricultural production trusteeship services can be divided into six categories: Agricultural material supply services, ploughing services, seeding services, land management services, harvesting services, and post-harvest services. Most farm households need three types of services, i.e., ploughing, seeding, and harvesting, and these three types of services are mechanical services. Therefore, in a further study on the effects of different trusteeship services on high-quality grain production, ploughing, seeding, and harvesting services are grouped in the same category of services. This paper will study the effect of agricultural material supply services, ploughing, seeding, and harvesting services, land management services, and post-harvest services on promoting high-quality grain production.

As shown in Table 6, agricultural material supply services have the best overall promotion effect on high-quality grain production, followed by post-harvest services, production management services, and ploughing, seeding, and harvesting services. Of these, agricultural material supply services are particularly effective in promoting the premiumization of grain production, with an *ATT* of 0.443, an increase of almost 160%. This may be related to the fact that agricultural material supply services can be a good promoter of the use of good seeds. The promotion effect of post-harvest services on the premiumization of grain production is also obvious, with an *ATT* of 0.29, an increase of nearly 60%. This may be due to the fact that post-harvest drying, storage, and agency services can prevent the post-harvest quality decline of grain. Production management services, such as unified prevention and control, are particularly effective at promoting the greenization of grain production, while ploughing, seeding, and harvesting services are more conducive to the promotion of premiumization and high efficiency of grain production.

**Table 6.** Promotion effects of different trusteeship services on high-quality grain production.

| Types of Trusteeship Services | | High-Quality Grain Production | High Efficiency of Grain Production | Premiumization of Grain Production | Greenization of Grain Production | Branding of Grain Production |
|---|---|---|---|---|---|---|
| Agricultural material supply | Experimental group | 0.614 | 0.618 | 0.701 | 0.482 | 0.619 |
| | Control group | 0.327 | 0.463 | 0.269 | 0.333 | 0.558 |
| | ATT | 0.287 *** | 0.155 *** | 0.433 *** | 0.149 *** | 0.06 *** |
| | | (0.015) | (0.01) | (0.022) | (0.021) | (0.013) |
| Ploughing, seeding and harvesting services | Experimental group | 0.558 | 0.613 | 0.597 | 0.467 | 0.63 |
| | Control group | 0.418 | 0.479 | 0.392 | 0.421 | 0.511 |
| | ATT | 0.139 *** | 0.134 *** | 0.205 *** | 0.046 *** | 0.119 *** |
| | | (0.025) | (0.015) | (0.039) | (0.03) | (0.015) |
| Production management services | Experimental group | 0.597 | 0.621 | 0.62 | 0.546 | 0.676 |
| | Control group | 0.439 | 0.516 | 0.479 | 0.337 | 0.514 |
| | ATT | 0.158 *** | 0.105 *** | 0.141 *** | 0.208 *** | 0.162 *** |
| | | (0.017) | (0.011) | (0.027) | (0.021) | (0.012) |
| Post-harvest services | Experimental group | 0.685 | 0.676 | 0.759 | 0.573 | 0.743 |
| | Control group | 0.455 | 0.536 | 0.47 | 0.388 | 0.549 |
| | ATT | 0.23 *** | 0.14 *** | 0.29 *** | 0.185 *** | 0.194 *** |
| | | (0.017) | (0.012) | (0.026) | (0.025) | (0.015) |

Note: The results of k-nearest neighbor matching (k = 4) are listed in this table, and the results by other matching methods are similar and so are not listed here. *** indicate that it is significant at the statistical level of 1%; standard deviations are shown in brackets.

The above conclusions are also strongly supported by previous relevant studies. For example, research by Vitor Henrique Vaz Mondo indicates that the use of good seeds can positively affect the yield and quality of corn in Pirabicaba of Brazil [34]; research by Rani Puthukulangara Ramachandran affirms that the application of grain storage techniques is very important to prevent grain deterioration [38]. Chunfang Yang and Zhong Ren also argue that integrated prevention and control services, precision fertilization (or soil testing formula fertilization), and the use of organic fertilizers are all beneficial to the green development of agriculture [10,11]. The study by Chen Tao shows that agricultural services are highly correlated with the high efficiency of agricultural production [7]. These studies confirm the contribution of individual agricultural production services to individual aspects of grain production. However, using the same sample, we can not only study the effects of individual services on individual aspects of the high-quality production of grain and their integrated effects but also compare the various effects. This is more helpful to find out what services are more conducive to the high-quality production of grain and can thus break through the dilemma that the results of previous studies are not suitable for comparison due to different samples.

### 4.4.2. Heterogeneity of Promotion Effects for Farm Households with Different Factor Endowments

Considering the promotion effect of agricultural production trusteeship for farm households with different factor endowments varies, the sample farm households are categorized into different groups based on the average value of the variables of the size of the household labor force and household types of agricultural machinery, and the sample of farm households is categorized into two groups: Those whose scale of operation is 10 mus or less and those whose scale of operation is more than 10 mus, based on the statistical standards for small farm households in China. According to data from the third agricultural census, China now has 230 million farm households, with an average household scale of operation of 7.8 mus, and of these farm households, 210 million are small-scale ones operating on less than 10 mus of cultivated land, accounting for more than 98% of agricultural operation entities. As is shown in Table 7, the promotion effect of agricultural production trusteeship on farm households with a small labor force is better than that of farm households with a large labor force; the promotion effect on small-scale farm households with less agricultural machinery is better than that on the large-scale farm households with more agricultural machinery. The above results may be explained by the fact that the farm households with a large labor force, a large scale of operation, or more types of agricultural machinery are mostly specialized grain growers. With a better resource endowment, specialized grain farmers enjoy an inherently higher level of high-quality production of grain. Moreover, small farm households, with a small agricultural labor force and a small scale of operation or capital, have an originally low level of high-quality production of grain. After the small farm households enter agricultural production trusteeship, the promotion effect of the trusteeship services on their high-quality production is remarkable. At present, Chinese grain producers are still primarily small farm households, so agricultural production trusteeships will continue to play an important role in boosting the high-quality production of grain in China.

The findings of this paper are highly consistent with previous related studies. Research by Siyu Yang et al. indicates that agricultural machinery service has a greater impact on the land productivity of small-scale farms [30]; Ma Li et al. even suggest that socialized services promote the organic integration of small farm households with modern agriculture, which can better motivate small farm households to engage in modern agricultural production [18]. Based on these, we carry out a more in-depth and systematic study to classify small farm households not only in terms of their scale operation but also in terms of their labor force and scale of capital, and then we compare the promotion effects of agricultural production trusteeship services on high-quality production of each type of farm households.

**Table 7.** Promotion effect of agricultural production trusteeship on high-quality grain production by trusteeship farmers with different factor endowments.

| Grouping Variables | | High-Quality Grain Production | High Efficiency of Grain Production | Premiumization of Grain Production | Greenization of Grain Production | Branding of Grain Production |
|---|---|---|---|---|---|---|
| Number of labor force | | | | | | |
| 3 persons or less | Experimental group | 0.62 | 0.662 | 0.655 | 0.544 | 0.665 |
| | Control group | 0.309 | 0.411 | 0.248 | 0.337 | 0.505 |
| | ATT | 0.311 *** | 0.25 *** | 0.408 *** | 0.207 *** | 0.16 *** |
| | | (0.026) | (0.014) | (0.043) | (0.035) | (0.016) |
| More than 3 persons | Experimental group | 0.62 | 0.645 | 0.732 | 0.441 | 0.623 |
| | Control group | 0.375 | 0.436 | 0.434 | 0.248 | 0.495 |
| | ATT | 0.245 *** | 0.209 *** | 0.298 *** | 0.193 *** | 0.127 *** |
| | | (0.039) | (0.019) | (0.068) | (0.049) | (0.029) |
| Scale of operation | | | | | | |
| 10 mus or less | Experimental group | 0.613 | 0.654 | 0.663 | 0.514 | 0.662 |
| | Control group | 0.297 | 0.406 | 0.252 | 0.296 | 0.484 |
| | ATT | 0.316 *** | 0.248 *** (0.014) | 0.411 *** | 0.218 *** | 0.178 *** |
| | | (0.026) | | (0.043) | (0.034) | (0.017) |
| More than 10 mus | Experimental group | 0.648 | 0.673 | 0.714 | 0.538 | 0.653 |
| | Control group | 0.385 | 0.446 | 0.381 | 0.352 | 0.52 |
| | ATT | 0.263 *** | 0.227 *** | 0.333 *** | 0.186 *** | 0.132 *** |
| | | (0.033) | (0.017) | (0.065) | (0.047) | (0.027) |
| Types of agricultural machinery | | | | | | |
| 1 type or less | Experimental group | 0.621 | 0.656 | 0.652 | 0.555 | 0.652 |
| | Control group | 0.328 | 0.426 | 0.277 | 0.346 | 0.277 |
| | ATT | 0.293 *** | 0.23 *** | 0.375 *** | 0.209 *** | 0.375 *** |
| | | (0.026) | (0.014) | (0.044) | (0.037) | (0.044) |
| More than 1 type | Experimental group | 0.619 | 0.664 | 0.717 | 0.449 | 0.646 |
| | Control group | 0.355 | 0.436 | 0.361 | 0.296 | 0.506 |
| | ATT | 0.264 *** | 0.229 *** | 0.355 *** | 0.153 *** | 0.139 *** |
| | | (0.028) | (0.014) | (0.046) | (0.036) | (0.02) |

Note: The results of k-nearest neighbor matching (k = 4) are listed in this table, while the results of other matching methods are similar and so are not listed here. *** indicate that it is significant at the statistical level of 1%; standard deviations are shown in brackets.

### 5. Conclusions and Insights

Based on research data from October to December 2020 in five major wheat-producing provinces, including Henan, Shandong, Anhui, Hebei, and Jiangsu provinces, we have investigated the promotion effect of agricultural production trusteeship on the high-quality production of grain of the trusteeship farm households using propensity score matching method. The research findings are as follows:

1.  Agricultural production trusteeships can promote the high-quality production of grain by promoting high efficiency, premiumization, greenization, and branding of grain production, with the promotion of premiumization being the most effective, followed by high efficiency, greenization, and branding.

2.  The promotion effects of different trusteeship services on the high-quality production of grain differ, with agricultural material supply services being the most effective, followed by post-production services, and then production management and ploughing, seeding, and harvesting services. Specifically, agricultural material supply services, ploughing, seeding, and harvesting services, and post-production services are the most effective at promoting the premiumization of grain production, and production management services have the most significant driving effect on the greenization of grain production.

3.  Compared with farm households with better resource endowments, agricultural production trusteeship has a better promotion effect on the high-quality production of grain by small-farm holders with a small labor force, a small scale of operation, and fewer types of agricultural machinery.

The above results show that agricultural production trusteeships can effectively promote high-quality grain production and help secure grain security in China. Hence, in order to promote the development of agricultural production trusteeship, the Chinese government should incorporate it into the grain security policy system, make it a priority support area for agriculture-related funds, and focus on improving the business level of trusteeship organizations. Meanwhile, given the differences in the effectiveness of different trusteeship services in promoting high-quality grain production, the government should focus on supporting trusteeship services with a significant promotion effect.

In addition, given the fact that agricultural production trusteeship has a better driving effect on small farm households and that small farm households are still the mainstay of grain cultivation in China, it is important to raise the awareness of agricultural production trusteeship of small farm households and guide more small farm households to choose agricultural production trusteeship. At the same time, attention should be paid to new agricultural operators with service potential and support them to provide trusteeship services for small farm households in the surrounding areas.

**Author Contributions:** Conceptualization, X.S. and Y.W.; methodology, X.S., Y.W. and X.L.; software, Y.W. and X.L.; validation, X.L.; formal analysis, J.L. and S.S.; investigation, X.S., Y.W. and X.L.; resources, J.L.; data curation, Y.W. and X.L.; writing—original draft preparation, X.S.; writing—review and editing, F.Z.; visualization, X.L.; supervision, X.S. and J.L.; project administration, X.S.; funding acquisition, X.S. All authors have read and agreed to the published version of the manuscript.

**Funding:** This research was funded by Shandong Provincial Office for Philosophy and Social Sciences, under the Key Project of Social Science Planning of Shandong Province, China, grant number 22BJJJ02.

**Data Availability Statement:** All data can be obtained by email from the corresponding author.

**Conflicts of Interest:** The authors declare no conflict of interest.

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
