# Peer review of "Promotion Effect of Agricultural Production Trusteeship on High-Quality Production of Grain—Evidence from the Perspective of Farm Households"

_agronomy, doi:10.3390/agronomy13082024_

Round 1
Reviewer 1 Report
Dear Authors
thanks for your paper that is clear and well written, and the topic is interesting.
I just suggest you some little adjustments:
1: in the introduction you should better focus the problem
2: in the theoretical framework highlight also problems
3: in conclusion and insights underling also the problems of truesteship
Author Response
Response to Reviewer 1 Comments
Dear Editor and Reviewer:
Thank you for giving us the opportunity to submit a revised copy of the manuscript. We appreciate the time and effort that you have dedicated to providing your constructive suggestions on this manuscript. Your comments and concerns are highly insightful, enabling us to improve the quality of the manuscript. All revisions to the manuscript have been marked up by using tracking changes function. We hope that this revised version has satisfactorily addressed all of your concerns. In the following, the point-by-point responses to each of the comments are presented.
The paper uses propensity score matching (PSM) to find out the promotion effect of agricultural production trusteeship on high-quality production of grain in China. The paper can be improved with the following issues addressed.
Point 1. in the introduction you should better focus the problem.
Response 1: Thank you very much for your suggestions. We completely agree with you, in the introductory section, we have focused more on the question " agricultural production trusteeship can contribute to high-quality grain production, which in turn can guarantee food security". The supplementary part is as follows:
At present, China's grain production is relatively safe in terms of quantity, but the quality of grain and the ecological environment are worrisome. (line36-37)
Hence, the purpose of this paper is to carry on an in-depth and systematic study of the promotion effect of agricultural production trusteeship on high-quality production of grain in multiple dimensions including high efficiency, premiumization, greenization and branding of grain production, so as to better accomplish the multidimensional goals of China's grain quantity security, quality security and ecological security. (line59-63)
Point 2. in the theoretical framework highlight also problems.
Response 2: Thank you very much for your professional work on the paper. In the theoretical framework section, we have further highlighted the themes and the additions are as follows:
High-quality grain production is the foundation for ensuring China's grain security in the new stage. And the whole-process, professional and large-scale agricultural production trusteeship services can effectively promote the high efficiency, premiumization, greenization and branding of grain production, and then solidly promote high-quality production of grain, and effectively guarantee the grain quantity, quality and ecological security of China from the production end. The specific framework for agricultural production trusteeship to promote high-quality production of grain is shown in Figure 1. (line139-145)
Point 3. in conclusion and insights underling also the problems of trusteeship.
Response 3: Thank you very much for your valuable suggestions, in conclusion and insights, we have focused more on the problems of agricultural production trusteeship services, as follows:
The above results show that agricultural production trusteeship can effectively promote high-quality production of grain and help secure grain security of China. Hence, in order to promote the development of agricultural production trusteeship, the Chinese government should incorporate it into the grain security policy system, make it a priority support area for agriculture-related funds, and focus on improving the business level of trusteeship organizations. Meanwhile, given the differences in the effectiveness of different trusteeship services in promoting high-quality production of grain, the government should focus on supporting the trusteeship services with significant promotion effect.
Besides, given the fact that agricultural production trusteeship has a better driving effect on small farm households, and that small farm households are still the mainstay of grain cultivation in China. Hence, it is important to raise the awareness of agricultural production trusteeship of the small farm households and to guide more small farm households to choose agricultural production trusteeship. At the same time, attention should be paid to exploring new agricultural operators with service potential and support them to provide trusteeship services for small farm households in the sur-rounding areas. (line582-596)

Reviewer 2 Report
Comments and suggestions:
Thanks for submitting the paper! Overall, the manuscript is well done, and brings some thing new to the research arena. Please check some of my comments and suggestions to improve the manuscript.
Abstract:
It brings the key findings of the study well.
I. Introduction: is well presented with good background information related to the study. Objectives are clearly presented with the needed arguments.
2. Literature review: overall bring enough facts and figures associated with the purpose of the study. All sub sections are OK. A bit more literature support given to 2.2.1, would do a better job (I mean the first 1-5 lines). Also, 2.2.3 last few points can be supported as well.
Figure 1: Do not see any mention in the text, when to refer that? Also, is that by the authors or taken from somewhere else? Better to give source.
3. Data sources, Research methods and variable selection
Overall good.
Page 5, under foot note 4, what does it means by “mus”? Seems like a unit or scale? Not clear?
I think you have pre-tested the questionnaire with a random group, and if so, please mention it somewhere. Also, how many trained enumerators collected data (I see authors say we did, but the reader would like to see a bit more in detail).
Methodology was presented well. Under 3.3.2 – households with and without trusteeship is based on the number of services, as more than 4 is “under” and less than 4 is not “under”. Not clear how the services number 4 was selected? A bit of explanation will be helpful to the reader.
4. Empirical results and analysis
Overall presented well. The tables and the graph are meaningful and explained adequately.
References: Authors have gone through many recent literatures, and that is good.
Author Response
Response to Reviewer 2 Comments
Dear Editor and Reviewer:
Thank you for giving us the opportunity to submit a revised copy of the manuscript. We appreciate the time and effort that you have dedicated to providing your constructive suggestions on this manuscript. Your comments and concerns are highly insightful, enabling us to improve the quality of the manuscript. All revisions to the manuscript have been marked up by using tracking changes function. We hope that this revised version has satisfactorily addressed all of your concerns. In the following, the point-by-point responses to each of the comments are presented.
The paper uses propensity score matching (PSM) to find out the promotion effect of agricultural production trusteeship on high-quality production of grain in China. The paper can be improved with the following issues addressed.
Point 1. Introduction: is well presented with good background information related to the study. Objectives are clearly presented with the needed arguments.
Response 1: Thank you very much for your valuable comments, your comments are a great inspiration.
Point 2. Literature review: overall bring enough facts and figures associated with the purpose of the study. All sub sections are OK. A bit more literature support given to 2.2.1, would do a better job (I mean the first 1-5 lines). Also, 2.2.3 last few points can be supported as well. Figure 1: Do not see any mention in the text, when to refer that? Also, is that by the authors or taken from somewhere else? Better to give source.
Response 2: Thank you very much for your professional work on the paper.
- Response about 2.2.1: The idea of " the use of good seeds of grain and intensive, standardized and scientific planting services are also conducive to the increase of grain yield " come from literature 33-35, which was accidentally deleted in the version of the paper submitted on June 3, and has been added again this time. Thank you for the reminder! (line154-156)
- Response about 2.2.3: In response to your suggestions, we have revised and improved this section, as follows:
Agricultural production trusteeship service usually adopts low-toxic and environment-friendly pesticides, fertilizers and other green inputs, and adopts green production technologies such as integrated prevention and control and straw return. These green production behaviors can not only directly reduce environmental pollution in the process of grain production, but also help improve farm households' green cognition level and improve their adoption of green production technologies, thereby effectively reduce the possible non-point source pollution caused by grain cultivation [41-42]. (line186-192)
- Response about Figure 1: Thanks for the suggestion, we have given a brief introduction to Figure 1, and in footnote 3 we have stated that the figure is produced by the authors of this paper, as well as mentioning it from its original position on line 144-145, as follows:
The specific framework for agricultural production trusteeship to promote high-quality production of grain is shown in Figure 1. (line144-145)
Point 3. Data sources, Research methods and variable selection
Overall good.
Page 5, under foot note 4, what does it means by “mus”? Seems like a unit or scale? Not clear? I think you have pre-tested the questionnaire with a random group, and if so, please mention it somewhere. Also, how many trained enumerators collected data (I see authors say we did, but the reader would like to see a bit more in detail). Methodology was presented well. Under 3.3.2 – households with and without trusteeship is based on the number of services, as more than 4 is “under” and less than 4 is not “under”. Not clear how the services number 4 was selected? A bit of explanation will be helpful to the reader.
Response 3: Thank you very much for your valuable comments and suggestions,
- Response about “mu”: In China, mu is the common statistical unit and official standard for land area (1 mu=666.67=1/5), and the explanation of mu was originally given in footnote 1 on page 2 of this paper, where the unit of mu first appears. In this revision, we have improved the original explanation there.
- Response about “I think you have pre-tested the questionnaire with a random group, and if so, please mention it somewhere”: We did conduct a pre-survey, which was not previously mentioned, and thank you very much for your reminder, we have added a description of it in this revision, as follows:
Before the formal investigation, our research team carried out a preliminary investigation in a number of typical administrative villages in Shandong Province. Through the pre-survey, we modified and improved the survey questionnaire, so as to determine the formal survey questionnaire. (line221-224)
- Response about “how many trained enumerators collected data”: Thank you for your suggestion, with regard to investigators, we have revised the details in the footnote 6 section on page 6:
Our investigation team consists of 25 people, including 5 teachers from Shandong University of Finance and Economics, 3 doctoral students, 12 master's students, and 5 undergraduates. The team was divided into five groups, each led by a teacher and responsible for the investigation in one province. The members of investigation team have received professional questionnaire training before conducting the formal survey, the training methods include practical training and conference training. We conducted practical training for the members participating in the pre-survey through pre-survey. In addition, after pre-survey, we conducted meeting training for all team members. Through training, team members are very familiar with the content of the questionnaire and have a deep understanding of the methods and other points for attention of the questionnaire.
- Response about “Not clear how the services number 4 was selected? A bit of explanation will be helpful to the reader.” Thank you very much for your suggestion. In the previous version, an explanation was given for this issue. Now we've refined it so that it's clearer and easier to understand, as follows:
Agricultural production trusteeship organizations can provide the above mentioned six types of services for grain production in a one-stop, all-inclusive manner. Since most farm households need four services including agricultural materials supply service, ploughing service, seeding service and harvesting service, regardless of whether they purchase agricultural production trusteeship services, in this paper, farm households who are under agricultural trusteeship service are defined as those who purchase four or more kinds of trusteeship services from the same trusteeship organization, while farm households who are not under agricultural production trusteeship are defined as those who purchase less than four kinds of services from the same trusteeship organization. (line349-358)
Point 4. Empirical results and analysis
Overall presented well. The tables and the graph are meaningful and explained adequately.
References: Authors have gone through many recent literatures, and that is good.
Response 4: Thank you for your comment, it is a great encouragement.
